# Recombinant cystatin-like protein-based competition ELISA for *Trichinella spiralis* antibody test in multihost sera

Yan Liu[1], Ning Xu[1], Yansong Li[1], Bin Tang[1], Hualin Yang[2], Weihua Gao[2], Mingyuan Liu[1]*, Xiaolei Liu[1]*, Yu Zhou[1,2]*

**1** Key Laboratory of Zoonosis Research, Ministry of Education, Institute of Zoonosis, College of Veterinary Medicine, Jilin University, Changchun, China, **2** College of Animal Sciences, Yangtze University, Jingzhou, China

☯ These authors contributed equally to this work.
* liumy36@163.com (ML); liuxlei@163.com (XL); zhouyurunye@sina.com (YZ)

**Data Availability Statement:** All relevant data are with the manuscript and its Supporting Information files.

## Abstract

### Objectives

*Trichinella spiralis* is a zoonotic parasite with a complex parasitic life cycle and exposed to animals or humans by infectious meat. To control transmissions of *T. spiralis* through the food chain to humans, sensitive and selective multihost sera-diagnosis is urgent needed for monitoring *T. spiralis* exposure.

### Methods

A competition enzyme-linked immunosorbent assay (cELISA) for *T. spiralis* infection diagnosis in multihost sera was developed based on recombinant cystatin-like protein (rCLP-cELISA) as well as monoclonal antibodies. The sensitivity and accuracy of the rCLP-cELISA were quantified using swine (n = 1316), mice (n = 189) and human (n = 157) serum samples. *T. spiralis*-antibody targeting test ability of the rCLP-cELISA in swine (n = 22) and human (n = 36), instead of other parasites or viruses antibodies, was evaluated.

### Results

The rCLP-cELISA showed high agreement with commercial ELISA kits in field swine sera assessed by Cohen's kappa value (κ = 0.7963). And it showed 100% specificity in human trichinellosis detection with sensitivity of 96.49%, no cross-reaction with other parasite or virus infections, and high positive detection rate of 87.5% in low-dose infected swine. Besides, the rCLP-cELISA exhibited potential in the detection of *T. spiralis*, *T. nelsoni* and *Trichinella* T8 infections.

### Conclusions

The rCLP-cELISA can be used for *T. spiralis*-associated antibody test in multihost sera.

**Funding:** ML was supported by the National Key Research and Development Program of China (2018YFC1602500, 2017YFC1601200), the National Natural Science Foundation of China (NO. 31520103916), Guangdong innovation and Enterpreneurial Research Team Program (NO. 2014ZT05S123), Jilin Provincial Science and Technology Development Project (20180520042JH), and Program for JLU Science and Technology Innovative Research Team (2017TD-32), http://www.nsfc.gov. http://www.jlkjxm.com; XL prof. was supported by the National Natural Science Foundation of China (NO. 31872467), http://www.service.most.gov.cn http://www.nsfc.gov.; YZ prof. was supported by the National Natural Science Foundation of China (NO. 31871888), http://www.nsfc.gov. The funders had no role in study design, data collection and analysis, decision to publish, or preparation of the manuscript.

**Competing interests:** The authors have declared that no competing interests exist.

## Author summary

Infections with *T. spiralis* that lives in host muscles for long periods of time are commonly difficult to diagnosis without causing a strong immune response or symptoms. The habit of eating raw/undercooked pork meat accidentally introduces humans into the exposure of *T. spiralis* that circulates between herds and rodents. There is an urgent need for serological antibody test of *T. spiralis* to monitor the infection of humans as well as hosts in the food chain, which is restricted by the mixture type of current used antigens and species-specific secondary antibodies for different hosts. We developed a novel recombinant cystatin-like protein-based competition enzyme-linked immunosorbent assay (rCLP-cELISA) employing monoclonal antibody. The proposed method showed considerable sensitivity and specificity in filed swine sera and human serum samples with good versatility in mice. Taking advantage of its controllable quality stability, the rCLP-cELISA had potential applications for screening of *T. spiralis* infections for multihost sera in one test. With the development of monoclonal antibody modification strategies and the discovery of antigenic proteins from parasitic pathogens, the proposed competition ELISA also provides useful reference for the improvement of serological assay for monitoring the exposure of zoonotic parasites.

## 1. Introduction

*Trichinella spiralis* is an important zoonotic parasite and commonly associated with domestic pigs with highly infective for humans [1–3]. *T. spiralis* is accounts for serious economic losses in the pig industry and hundreds of human infections worldwide annually [2–4]. Domestic pork is the main source of human trichinellosis, which results from consumer ingestion of infectious meat [5,6]. In China, the southwestern, central and northeastern parts are the main endemic areas of *T. spiralis* infections in herds, which is consistent with the highest prevalence of human trichinellosis reported in Yunnan, Hubei and Henan provinces [7,8]. In these endemic provinces, the habit of eating raw/undercooked pork meat is common, which accidentally introduces humans into the exposure. The potentially high rate of infection in domestic herds is responsible for the threat of human infection by *T. spiralis*, therefore, monitoring the exposure of herds, vectors and populations is beneficial for *T. spiralis* control and public health.

Parasites of the genus *Trichinella* are transmitted among hosts by the ingestion of infected muscles revealing two epidemiological cycles: domestic and sylvatic [9,10]. The domestic cycle occurs particularly between domestic swine and rodents [9–11]. With the presence of illegal feeding by left-over food containing *Trichinella* larvae, an important way of introducing the parasite to swine is transmission by vectors, such as rodents, in pig farms without proper control and presenting low levels of sanitary conditions in rural areas [9,11]. Considering the growing scale of meat market of rodents [12,13] as the vector of *T. spiralis* transmission, as well as high prevalence of the infection in domestic swine [3,7,8], it is important to focus on epidemiological investigations of hosts in the food chain for risk management and control of this food-born parasite.

Serological assays such as enzyme-linked immunosorbent assay (ELISA) have been recommended by the International Commission on Trichinellosis (ICT) for surveillance of *T. spiralis* infections in herds to ensure food safety [14,15]. ELISA is also the first recommendation for screening of human trichinellosis, since clinical symptoms are nonspecific and resemble those

of influenza or other disorders [15,16]. A number of ELISAs have been established for monitoring *T. spiralis* infections in animal hosts [17,18] or human host [19], respectively. Most of the assays are indirect ELISAs employing excretory-secretary (ES) products (ES-iELISA), a mixture-type antigens from muscle larvae collected strictly by living animals, which possess defects in reproducibility, quality control, high cost and cross-reactivity [17–19]. To control transmissions of *T. spiralis* through the food chain to humans, sensitive and selective multihost sera-diagnosis is urgent needed based on definitive antigenic components of ES products [18,20]. To date, only one multihost universal test has been reported [21] for the detection of *T. spiralis* infection, in which the mixed antigens were employed.

In our previous study [22,23], an antigenic cystatin-like protein (CLP) was proved to be expressed in various developmental stages of *T. spiralis*, and monoclonal antibodies (MAbs) against CLP were prepared [23]. In this study, a recombinant CLP (rCLP)-based competition ELISA (rCLP-cELISA) employing MAb for the detection of *T. spiralis* antibodies was proposed and evaluated using multiple host serum samples. Compared with commercial ELISA kits (ES-iELISAs, swine and human), the proposed rCLP-cELISA showed similar sensitivity and no cross-reactions with other parasite or virus infections in swine and human sera.

## 2. Materials and methods

### Ethics statement

Human and animal serum samples, as well as different species/genotypes of *Trichinella* larvae, were collected and conserved by the OIE Collaborating Centre for Food-Borne Parasites from the Asian-Pacific Region, Jilin University. All experiments in this study were approved by the Ethical Committee of Jilin University, China (permit number: 20170318 for animal serum samples; 2019-H-K13 for human sera). All the human serum samples were collected from adults, and the written informed consent was acquired from the adults before samples were used. The authors confirm that the ethical policies of this study, as noted on the journal's author guidelines page, have been adhered to ethical approvals as this report solely contains diagnostic samples taken in the field.

### 2.1 Parasite and sera

A total of 1720 serum samples used in this study were shown in **Table 1**. Negative swine sera were reconfirmed with a commercial ELISA kit (Qiagen, Cat. 273501), the corresponding diaphragm tissues of which were tested by artificial digestion [24]. Positive sera from swine and mice infected with *T. spiralis* (iss 534) were collected and stored at -80˚C. Positive human sera were collected from two family infection cases in which pork was privately raised and slaughtered in Yunnan province. Serum samples from 100 healthy donors from the nearby villages of the two cases were reconfirmed with a commercial ELISA kit (Abcam, ab108780). Sera from patients with *Clonorchis sinensis* were collected in Jilin province. Sera from patients with enterovirus 71 (EV71), coxsackievirus A16 (CA16) and other parasites infections were provided by National Institute of Parasitic Diseases of Chinese Centre for Disease Control and Prevention.

### 2.2 Immunoassay procedure of the rCLP-cELISA and cut-off values

The MAb 1H9 binding to native epitope [39] HEALFSSDLKQESGV [53] of CLP (GenBank: ABY60755.1) was selected using competitive ELISA (S1 Text) and was labelled with biotin by Sangon Biotech (Shanghai) Co., Ltd. Briefly, 96-well ELISA plates were coated with 1.25 µg/mL rCLP antigens in 100 µL per well. The plates were blocked with PBST containing 1% BSA at 37˚C for 1 h. MAb 1H9 with biotin conjugation (0.3 µg/mL) was prepared in 0.9%

**Table 1. Sera from different hosts with *Trichinella* or other pathogen infection used in this study.**

| Test groups | | Infected with | NO examined | Doses [a] | Dpi [b] |
|---|---|---|---|---|---|
| Swine | Cut-off value test | Digest negative [c] | 270 | - [d] | - |
| | Antibody kinetics test | *T. spiralis* | 108 | 200, 400, 600 | 7–120 |
| | Accuracy test | *T. spiralis* | 18 | 50, 100, 200 | 120 |
| | Field serum samples | - | 920 | - | - |
| | Cross-reactivity | *Toxoplasma gondii* | 3 | 400 | 90 |
| | | *Cryptosporidium parvum* | 2 | 400 | 90 |
| | | *Taenia solium* | 3 | 400 | 90 |
| | | *Ascaris suum* | 3 | 400 | 60 |
| | | *Trichuris suis* | 5 | 400 | 90 |
| | | Virus vaccine [e] | 6 | - | - |
| Mice | Cut-off value test | Digest negative [f] | 30 | - | - |
| | Antibody kinetics test | *T. spiralis* | 108 | 10, 20, 200, 400 | 3–60 |
| | Cross-species test | 12 species/genotypes of *Trichinella* [g] | 51 | 200 | 60 |
| Human | Cut-off value test | *Trichinella* [h] | 57 | Unknown | Unknown |
| | | Healthy donors [i] | 100 | - | - |
| | Cross-reactivity [j] | *Ascaris lumbricoides* | 3 | Unknown | Unknown |
| | | *Trichuris trichiura* | 4 | Unknown | Unknown |
| | | *Ancylostoma duodenale* | 3 | Unknown | Unknown |
| | | *Toxoplasma gondii* | 3 | Unknown | Unknown |
| Human | Cross-reactivity [j] | *Paragonimus* | 2 | Unknown | Unknown |
| | | *Fascioloa hepatica* | 3 | Unknown | Unknown |
| | | *Schistosoma japonicum* | 2 | Unknown | Unknown |
| | | *Clonorchis sinensis* | 8 | Unknown | Unknown |
| | | *Cerebral cysticercus* | 2 | Unknown | Unknown |
| | | Enterovirus | 6 | Unknown | Unknown |

No for number.

[a] Doses: larval inoculation dose in host.

[b] Dpi: days post infection.

[c] The whole diaphragm tissue was tested and serological tested negative using ES-iELISA (Qiagen).

[d] No test.

[e] Virus vaccines included porcine pseudorabies virus, porcine reproductive and respiratory syndrome virus, porcine circovirus, classical swine fever virus and foot-and-mouth disease virus.

[f] The whole muscle tissues were tested.

[g] 17 isolates belonging to 12 species/genotypes of *Trichinella* were tested including *Trichinella spiralis* (T1, iss 534), *Trichinella spiralis* (T1, iss 533), *Trichinella spiralis* (T1, iss 4), *Trichinella nativa* (T2, iss 70), *Trichinella britovi* (T3, iss 235), *Trichinella pseudospiralis* (T4, iss 141), *Trichinella pseudospiralis* (T4, iss 13), *Trichinella pseudospiralis* (T4, iss 470), *Trichinella murrelli* (T5, iss 35), *Trichinella murrelli* (T5, iss 415), *Trichinella* T6 (T6, iss 34), *Trichinella nelsoni* (T7, iss 37), *Trichinella* T8 (T8, iss 124), *Trichinella* T9 (T9, iss 408), *Trichinella papuae* (T10, iss 572), *Trichinella zimbabwensis* (T11, iss 1029) and *Trichinella patagoniensis* (T12, iss 1826).

[h] Trichinellosis pork consumers in two outbreaks tested by ES-iELISA (Abcam) received deworming treatments.

[i] Healthy donors from nearby villages of the outbreaks were tested by ES-iELISA (Abcam).

[j] Patients with *Clonorchis sinensis* infections received fecal microscopy and deworming therapy. Other samples were provided by National Institute of Parasitic Diseases of Chinese Centre for Disease Control and Prevention.

NaCL solution (w/v). Equal volumes of the serum (50 μL) and the MAb solution (50 μL) were dispensed into wells and incubated at 37˚C for 1 h. Unbound serum and/or MAb were removed by washing steps. Then, HRP-labelled avidin (Invitrogen, LOT 2197902) was diluted 1:500 in PBST containing 1% BSA, added to wells and incubated at 37˚C for 30 min. After three washing steps, colour was developed for 8 min using TMB substrate solution (Tiangen

Biotech Co., Ltd.). After stopping the reaction with 0.2 M $H_2SO_4$, the optical density (OD) of each well was read at 450 nm with a microplate reader.

The result for each sample was expressed as the competitive inhibition (PI) ratio [21]. The PI ratio was defined as follows: PI = $(1 - OD_{sample}/OD_{MAb}) \times 100\%$. The cut-off value in rCLP-cELISA was calculated based on the mean of negative sera plus two standard deviations (SD) (swine, n = 270; mice, n = 30) or the receiver operator characteristics curves (ROC) analysis (human, n = 157) [21,25,26].

In ES-iELISAs, the signal-to-positive (S/P) ratio, cut-off values in swine sera, the $OD_{450nm}$ values of human sera as well as controls (negative, cut-off and positive controls) were all calculated according to the manufacturer's instructions.

## 2.3 Evaluation of the rCLP-cELISA performance

**2.3.1 Analysis of antibody kinetics using *T. spiralis* infected swine sera.**   Antibody kinetics of *T. spiralis* infection in swine at various stages were determined using rCLP-cELISA and ES-iELISA (Qiagen). Serum samples were obtained from three different infection doses (200, 400 and 600 larvae) at different days post infection (dpi). Seroconversion was defined as the time point when the concentration of *T. spiralis* antibody caused a PI ratio or S/P ratio that exceeded the cut-off value.

**2.3.2 Accuracy test using *T. spiralis* infected swine sera.**   The accuracy of the rCLP-cELISA was evaluated by analyzing sera from swine infected with three different doses of *T. spiralis* including 50, 100 and 200 larvae. The results were compared with those obtained by ES-iELISA (Qiagen) using the same serum samples. According to the recommendation of the ICT [14,24], diaphragm tissue of individual pig was detected with artificial digestion to calculate the mean larvae per gram (LPG) of muscle in order to confirm the larvae burden. Serum samples (n = 920) collected from backyard pigs from Hubei province were used to assess the sensitivity and specificity [27–29] of the rCLP-cELISA in the field, corresponding diaphragm tissues (100 g per pig) were detected to confirm the infection status of each pig.

**2.3.3 The rCLP-cELISA validation in mice sera.**   Antibody kinetics of *T. spiralis* infection in mice at various stages were determined using rCLP-cELISA. Serum samples were collected from four different infection doses (10, 20, 200 and 400 larvae) on different infection days.

Serum samples of mice infected with 17 isolates belonging to 12 species/genotypes of *Trichinella* were collected. The PI ratio of each serum sample was compared with the threshold to evaluate the potential suitability of the rCLP-cELISA used in sera infected with different species of *Trichinella* larvae. These samples were reconfirmed by crude worm extract-western blot (CWE-WB) using corresponding larvae and MAb 1H9 according to previous studies [23].

**2.3.4 The rCLP-cELISA validation in human sera.**   The developed rCLP-cELISA was validated by analyzing 100 healthy donors sera and 57 sera from trichinellosis patients using ROC analysis. Six of these positive sera with different $OD_{450nm}$ values as well as different PI ratios were analyzed by western blot employing ES products (ES-WB) of *T. spiralis* (iss 534). ES products were produced as described in a previous study [22], and the procedure for ES-WB was performed as published protocols [30,31]. Two serological negative sera coming from the nearby villages of trichinellosis patients were used as controls.

**2.3.5 Serological cross-reactivity validation of the rCLP-cELISA.**   The cross-reactivity of the rCLP-cELISA was evaluated using sera of other parasite infections and virus immunizations/infections in swine and human hosts, which compared with that of ELISA kit. According to the recommendation of the ICT [14,15], serological positive samples by ELISAs were reconfirmed with ES-WB, as well as the western blot employing rCLP (rCLP-WB). The procedure for rCLP-WB was performed as our previous study [23].

### 2.4 Statistical analysis

The results were expressed as the mean ± SD of the antibody kinetics of *T. spiralis* infection in swine and mice. Other results were expressed as the mean. Statistical analysis was performed by GraphPad Prism 6 software for Windows.

## 3. Results

### 3.1 Analytical performance of the rCLP-cELISA in swine sera

The MAb 1H9, with highest negative-to-positive ratio among three MAbs, was suitable for the development of rCLP-cELISA (S1 Text). The cut-off value of the rCLP-cELISA was determined to be 52% (S1 Fig). Antibody dynamics were determined in three different dose groups with rCLP-cELISA and ES-iELISA (**Fig 1**). The continuous serological positivity detected by rCLP-cELISA occurred later than that of ES-iELISA in the 600 and 400 larvae dose groups, in which seroconversion for continuous serological positivity delayed one sampling time point of 5 (600 larvae, **Fig 1A** and **1D**) or 15 (400 larvae, **Fig 1B** and **1E**) days. This delay time in seroconversion using rCLP-cELISA cannot be determined accurately due to the setting interval of sampling time points. When the infective dose was reduced from 600 to 200 larvae per pig, seroconversion for continuous serological positivity was delayed from 35 to 90 dpi in rCLP-cELISA and from 30 to 90 dpi in ES-iELISA (**Fig 1A, 1C, 1D** and **1F**). The PI ratios exceeded the threshold at 17 dpi and 19 dpi (**Fig 1D** and **1E**). A similar pattern of antibody dynamics was detected in the 200 larvae group, although PI ratios did not exceed the cut-off value (**Fig 1F**).

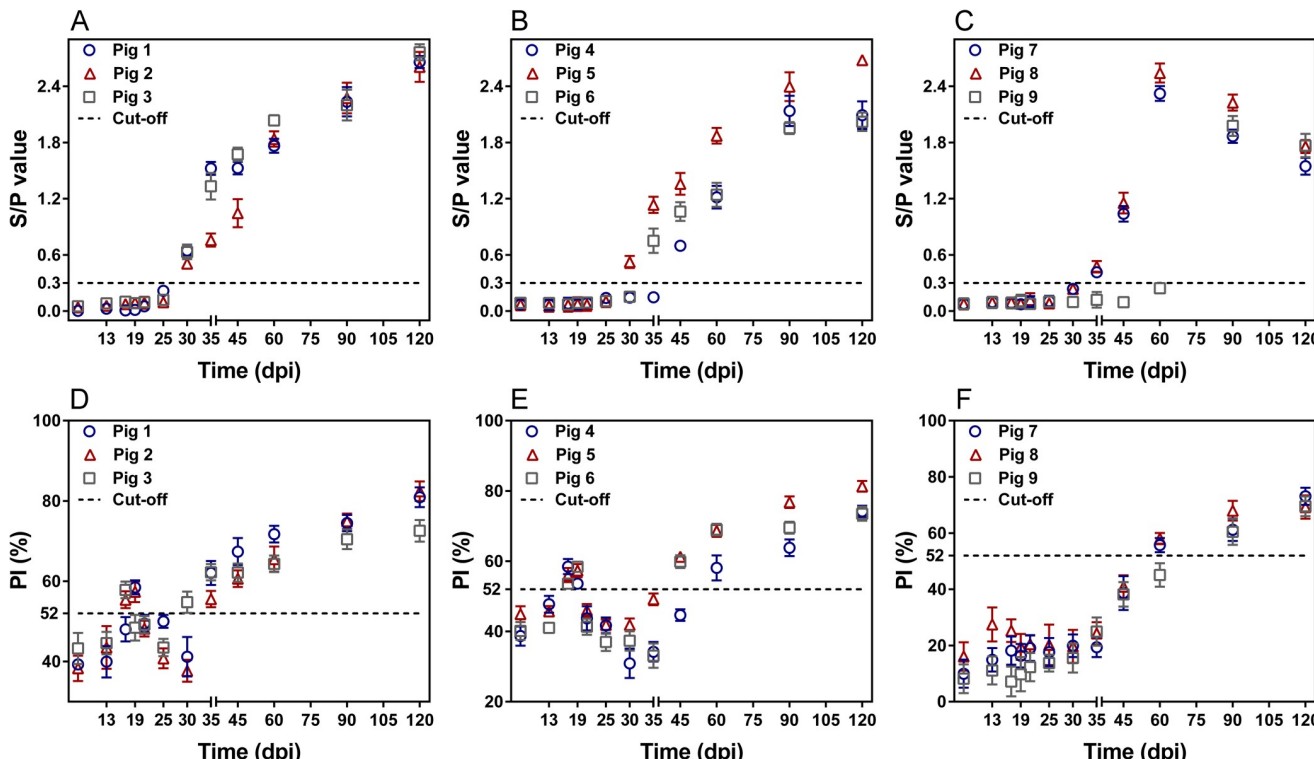

**Fig 1. Kinetics of competitive inhibition (PI) ratios of infected swine detected by rCLP-cELISA compared with ES-iELISA.** Kinetics of signal-to-positive (S/P) values in the ES-iELISA (A, B, C) or PI ratios in the rCLP-cELISA (D, E, F) of serum samples from swine experimentally infected with 600 larvae (A, D), 400 larvae (B, E) or 200 larvae (C, F) of *Trichinella spiralis*.

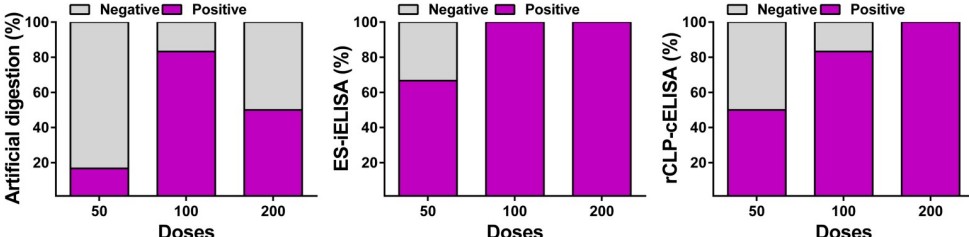

**Fig 2. Accuracy testing of the rCLP-ELISA in low-dose *Trichinella spiralis* infected swine compared with ES-iELISA and artificial digestion.** Swine (n = 18) were infected with *Trichinella spiralis* at three low doses (200, 100 and 50 larvae per pig). Serum samples were tested at 120 days post infection (dpi), and the corresponding diaphragm tissues were detected with artificial digestion. The results were displayed as the percentages of swine that scored positive or negative as determined by the three techniques.

The accuracy of rCLP-cELISA was evaluated through a recovery test (**Fig 2**), analyzing the matrix (sera) spiked samples of the low-dose groups including 50, 100 and 200 larvae. The results of rCLP-cELISA were compared with those detected by ES-iELISA. All pigs were examined by artificial digestion to calculate the mean larvae burdens (S1 Table). In 200 larvae group, the mean larvae burden of the diaphragm was less than 0.50 LPG, and the 100% positive rate of rCLP-cELISA was the same as that of ES-iELISA. When the mean larvae burden reduced to 0.025 LPG, 5/6 pigs were serologically positive using rCLP-cELISA. Since there were 4 positive pigs using ES-iELISA in the 50 larvae group, 3/4 pigs tested positive with rCLP-cELISA when the larvae burden of the diaphragm was as low as 0.005 LPG.

The agreement between rCLP-cELISA and ES-iELISA in field swine sera (n = 920) was assessed by Cohen's κ analysis (**Table 2**). According to κ values reported previously [27], a substantial agreement (κ = 0.7963) was observed between the two methods. In **Table 3**, field samples were classified as true positive or negative according to their larvae burden in diaphragm. The results showed that the proposed rCLP-cELISA are high sensitivity (*se*: 1.0000) with specificity of 0.9956.

## 3.2 Evaluation of the rCLP-cELISA using infected mice sera

The cut-off value of the rCLP-cELISA in mice sera was 39% (S2 Fig). In the different infection dose groups of mice (**Fig 3**), PI ratios exceeded the threshold at 14 dpi followed by a decrease, and again exceeded the cut-off value at 35 dpi (10 larvae), 42 dpi (20 larvae), 30 dpi (200 larvae)

**Table 2. Cohen's Kappa Statistic for measuring the agreement between rCLP-cELISA and ES-iELISA in field serum samples of swine.**

| Classified by ES-iELISA | Classified by rCLP-cELISA | | Total |
|---|---|---|---|
| | $T_+$ [a] | $T_-$ [b] | |
| $D_+$ [c] | 4 | 1 | 5 |
| $D_-$ [d] | 1 | 914 | 915 |
| Total | 5 | 915 | 920 |
| κ [e] | 0.7963 | | |

[a] Samples were serological tested positive using rCLP-cELISA.

[b] Samples were serological tested negative using rCLP-cELISA.

[c] Samples were serological tested positive using ES-iELISA (Qiagen).

[d] Samples were serological tested negative using ES-iELISA (Qiagen).

[e] The κ value was interpreted according to the Landis and Koch descriptors [27].

**Table 3. Diagnostic performance of the rCLP-cELISA in field serum samples of swine.**

| Classified by artificial digestion (as the gold standard) | Classified by rCLP-cELISA | | Total |
|---|---|---|---|
| | $T_+$ [a] | $T_-$ [b] | |
| $D_+$ [c] | 1 | 0 | 1 |
| $D_-$ [d] | 4 | 915 | 919 |
| Total | 5 | 915 | 920 |
| *se* [e] | 1.0000 | | |
| *sp* [e] | 0.9956 | | |
| Ĵ [e] | 0.9956 | | |

[a] Samples were serological tested positive using rCLP-cELISA.

[b] Samples were serological tested negative using rCLP-cELISA.

[c] Diaphragm tissue samples (100 g per pig) were tested positive using artificial digestion.

[d] Diaphragm tissue samples (100 g per pig) were tested negative using artificial digestion.

[e] The three parameters sensitivity (*se*), specificity (*sp*) and Youden index (Ĵ) were calculated according to the Youden descriptor [28].

and 21 dpi (400 larvae). The pattern of antibody dynamics detected in mice was similar to that in swine, regardless of whether the dose of infection was high (200 and 400 larvae) or low (10 and 20 larvae).

Serum samples of mice infected with different species/genotypes of *Trichinella* were tested with rCLP-cELISA at 60 dpi. As shown in **Fig 4A**, PI ratios of sera from 5 isolates were significantly above the threshold, including *T. spiralis* (T1, iss 534, iss 4 and iss 533), *T. nelsoni* (T7, iss 37) and *Trichinella* T8 (iss 124), which reflected the diagnostic potential of rCLP-cELISA used in these isolates of *Trichinella* infections. The CWE-WB was performed to confirm the presence of the epitope identified by MAb 1H9 (**Fig 4B**). Consistent with the results of rCLP-cELISA, MAb 1H9 can identify different patterns of bands specifically with the crude antigens of corresponding isolates. For *T. spiralis* (T1) and *T. nelsoni* (T7), characteristic bands were found at about 45 kDa and 2 or 3 bands at 25–35 kDa. The band at about 45 kDa of *Trichinella* T8 (iss 124) was higher than that of the two species with no band at 25–35 kDa. Although similar bands were found in the other 12 isolates, their corresponding sera were not serologically

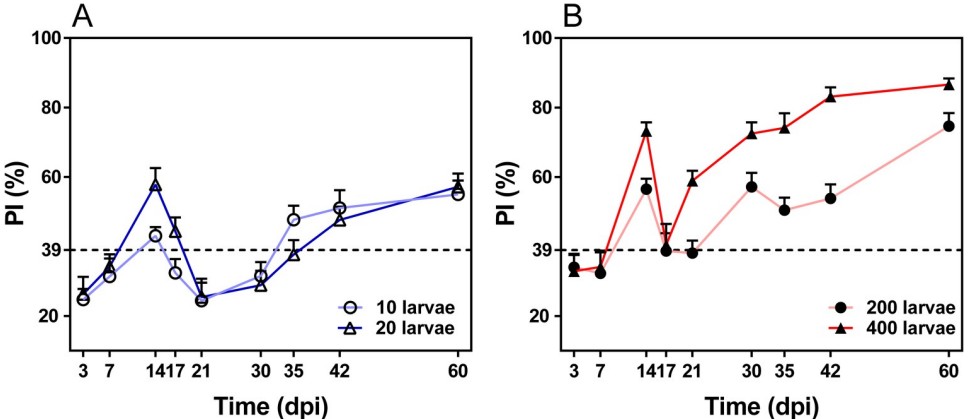

**Fig 3. Antibody kinetics of *Trichinella spiralis* infection in mice determined by rCLP-cELISA.** The cut-off value represented by the dotted line was evaluated on the basis of PI ratios (S3 Fig). Kinetics of PI ratios in the rCLP-cELISA of serum samples from mice experimentally infected with 10 larvae (A), 20 larvae (A), 200 larvae (B) or 400 larvae (B) of *Trichinella spiralis*.

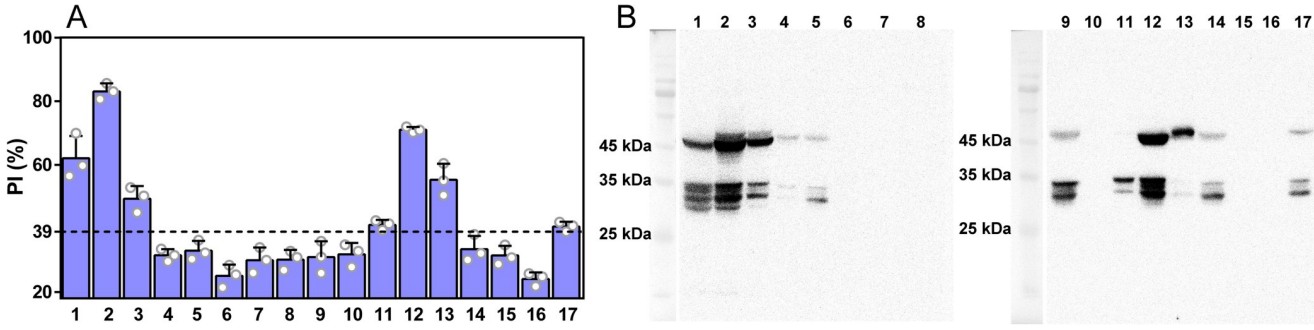

**Fig 4. Preliminary validation of the rCLP-cELISA in mice infected with different species/genotypes of *Trichinella*.** A. PI ratios of serum samples of mice infected with 17 isolates of *Trichinella*. The cut-off value represented by the dotted line. The group value was the mean ± SD of three independent experiments (n = 3). B. Analyses of western blot using crude worm extract antigens of 17 isolates of *Trichinella*. Groups 1–17 (A) and lanes 1–17 (B) represented 17 isolates of *Trichinella* including *Trichinella spiralis* (T1, iss 534) (group 1, lane 1), *Trichinella spiralis* (T1, iss 533) (group 2, lane 2), *Trichinella spiralis* (T1, iss 4) (group 3, lane 3), *Trichinella nativa* (T2, iss 70) (group 4, lane 4), *Trichinella britovi* (T3, iss 235) (group 5, lane 5), *Trichinella pseudospiralis* (T4, iss 141) (group 6, lane 6), *Trichinella pseudospiralis* (T4, iss 13) (group 7, lane 7), *Trichinella pseudospiralis* (T4, iss 470) (group 8, lane 8), *Trichinella murrelli* (T5, iss 35) (group 9, lane 9), *Trichinella murrelli* (T5, iss 415) (group 10, lane 10), *Trichinella* T6 (T6, iss 34) (group 11, lane 11), *Trichinella nelsoni* (T7, iss 37) (group 12, lane 12), *Trichinella* T8 (T8, iss 124) (group 13, lane 13), *Trichinella* T9 (T9, iss 408) (group 14, lane 14), *Trichinella papuae* (T10, iss 572) (group 15, lane 15), *Trichinella zimbabwensis* (T11, iss 1029) (group 16, lane 16) and *Trichinella patagoniensis* (T12, iss 1826) (group 17, lane 17).

positive according to rCLP-cELISA. This result may be related to the difference in antigenicity of the corresponding protein containing the epitope in different isolates, or the difference in competitiveness of serum antibody with 1H9 for the epitope.

## 3.3 Preliminary validation of the rCLP-cELISA in humans

The developed rCLP-cELISA distinguished well between 57 sera of trichinellosis patients and 100 sera of healthy donors (**Fig 5A**). According to the ROC analysis (**Fig 5B**), the threshold was set as 40.72% when the specificity reached 100% with sensitivity of 96.49%. The area under the curve (AUC) of PI was 0.9981, showing that the rCLP-cELISA are highly accurate (0.9 < AUC < 1.0) [25,26]. This result indicated that the defined cut-off value was suitable for evaluating rCLP-cELISA result to determine the presence/absence of *Trichinella* antibodies in human serum. In **Fig 5C**, each serum sample of trichinellosis patient was provided two values, one recorded with ES-iELISA and the other one recorded with rCLP-cELISA. The correlation

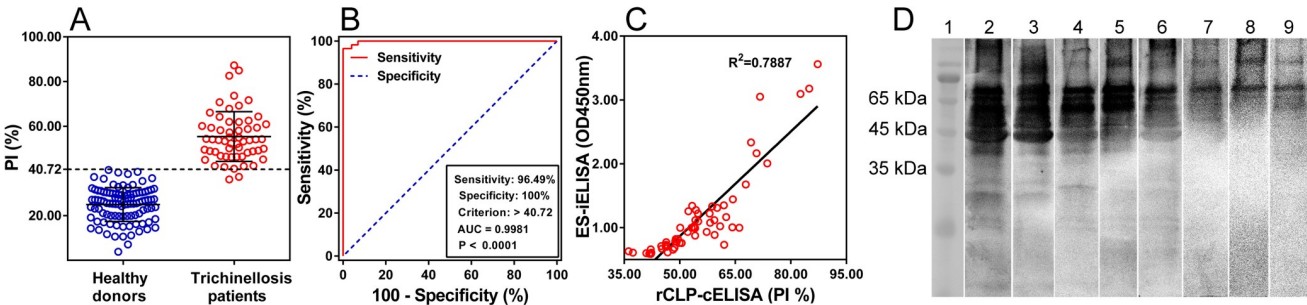

**Fig 5. Validation of the rCLP-cELISA in human sera.** A. PI ratios in rCLP-cELISA using 100 healthy donor' sera and 57 *Trichinella*-positive sera. B. Receiver operator characteristic curves (ROC) analysis using PI ratios of 157 human serum samples. The optimal cut-off value and area under the curve, determined for PI, were 40.72% and 0.9981, respectively. C. Scatter plots show the correlation between the rCLP-cELISA and ES-iELISA ($R^2$ = 0.7887). D. Confirmatory analysis of select participant samples using ES products-western blot. Lane 1: molecular weight markers; Lanes 2–9: Six *Trichinella*-positive sera and two negative sera determined by both rCLP-cELISA and ES-iELISA (cut-off value was set at 0.58 optical density value in this plate) (Abcam, UK). Their optical density values in ES-iELISA were 3.56, 3.05, 1.35, 1.31, 0.76, 0.70, 0.42, and 0.39. The corresponding PI ratios in rCLP-cELISA were 87.25%, 71.67%, 61.92%, 60.11%, 50.51%, 46.42%, 36.75%, and 29.67%, respectively.

between the two ELISAs was intense ($R^2$ = 0.7887) [29], suggesting a direct relationship between OD scores in ES-iELISA and PI ratios in rCLP-cELISA in human sera. A three-band pattern at 45–65 kDa [16,30,32] was determined in the ES-WB test using positive sera (**Fig 5D** lanes 2–6, black box). With PI ratios as well as OD values of positive sera decreased, the three-band gradually weakened, revealing the high consistency among the three methods. However, the characteristic pattern of sera with weak antibody positivity was not significantly distinguishable from those of antibody-negative samples (**Fig 5D** lanes 7–9). The results showed that the rCLP-cELISA exhibited higher sensitivity than traditional ES-WB in screening human trichinellosis [14–16].

### 3.4 Cross-reactivity of the rCLP-cELISA

All serum samples from swine infected with other parasites or immunized with virus vaccines were serologically negative by rCLP-cELISA and ES-iELISA (**Fig 6A** and **6B**). Compared with the rCLP-cELISA, cross-reaction occurred in human sera in two patients with *Clonorchis sinensis* using ES-iELISA (**Figs 6C and 6D,** and S3), although $OD_{450nm}$ values of other six samples in *Clonorchis sinensis* group did not exceed the threshold (S3 Fig). The two samples showing cross-reaction as well as one sample that slightly lowered the threshold were detected by ES-WB compared with rCLP-WB. Since the molecular weight of rCLP was about 47 kDa [23], no specific bands were tested using corresponding sera (**Fig 6E**). But the bands of the two

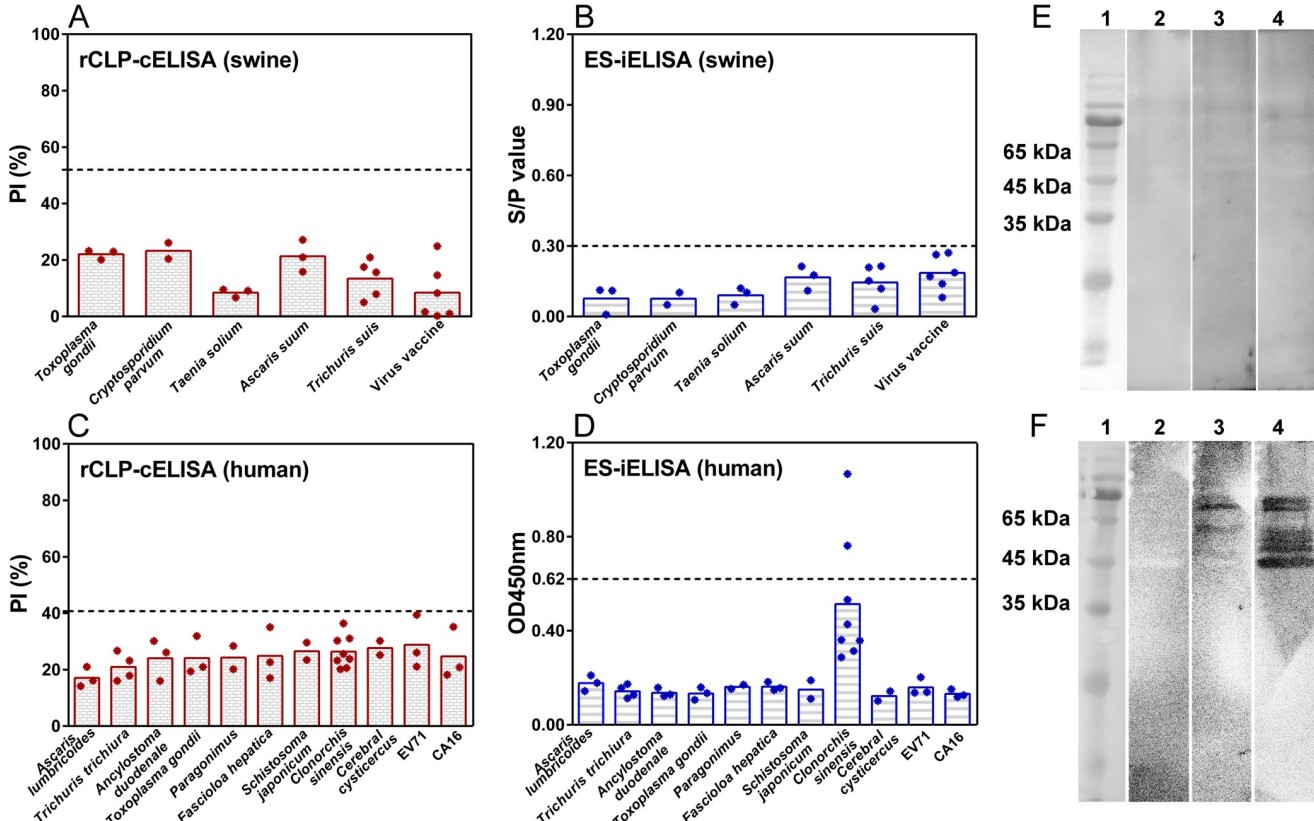

**Fig 6. Cross-reactivity validation of the rCLP-cELISA in swine and human.** Swine sera from other parasite (n = 16) infections and virus vaccine immunizations (n = 6) were tested by rCLP-cELISA (A) and ES-iELISA (B). Human serum samples from other parasite (n = 30) and enterovirus (n = 6) infections were tested by rCLP-cELISA (C) and ES-iELISA (D). The rCLP-western blot analysis (E) using selected sera of patients with *Clonorchis sinensis* compared with ES products-western blot test (F). Lane 1: molecular weight markers; Lane 2: Serum of patient with *Clonorchis sinensis* tested negative by both rCLP-cELISA and ES-iELISA; Lanes 3 and 4: Serum of patients with *Clonorchis sinensis* tested negative by rCLP-cELISA but positive by ES-iELISA.

samples at 45–65 kDa were tested in ES-WB (**Fig 6F** lanes 3–4), which were partly consistent with the three-band pattern of the positive reference sera (**Fig 5D** lanes 2–7). In addition, the reactivity of antibody-negative samples tested by ES-iELISA in the two regions (Yunnan and Jilin provinces) were completely different (**Figs 5D** lanes 8–9 and **6F** lane 2). The results indicated that rCLP-cELISA exhibited higher selectivity than ES-iELISA in patients with *Clonorchis sinensis*.

## 4. Discussion

ES products, as a mixture of multiple proteins (S4 Fig), require appropriate culture conditions and suitable purification methods to ensure the specificity of the detection results [17–19]. In the case of that every component of ES products is unknown, inter-batch and intra-batch quality controls of ES production are difficult. In addition to the differences in species/genotypes of *Trichinella* and in host species, the seroconversion time point of swine with the same infection dose varied greatly (S2 Table), which was largely related to the quality differences in ES products among different laboratories.

Our previous studies have shown that indirect ELISA employing rCLP can detect specific antibodies in swine sera collected from pigs infected with high dose ($\geq$ 1000 larvae per pig) of *T. spiralis* [23]. MAb 1H9 against epitope [39] HEALFSSDLKQESGV [53] of CLP was evaluated and found to be useful for establishing a competition ELISA. Detection methods based on specific antigens and MAbs can be beneficial to improve sera-diagnosis of *T. spiralis* infection, taking advantage of its controllable quality, low cost, good stability and reproducibility. The limit of detection (LOD) of artificial digestion was 1.0 LPG, the result of which was not stable when sampled near the LOD due to the limited sample size of tissue and personnel operation factors [24,33]. According to previous studies (S2 and S3 Tables), we focused on the sensitivity of rCLP-cELISA at a low infective dose ($<$ 800 larvae per pig). The rCLP-cELISA showed a 100% positive detection rate when the larvae burden was lower than 1.0 LPG as well as 66.7% when the larvae burden was even lower than 0.03 LPG. And it had a good agreement rate with ES-iELISA in field samples ($\kappa$ = 0.7963). Results of field samples testing showed that rCLP-cELISA had considerable sensitivity (*se* = 1.0000) with good application prospects for monitoring swine infection status to ensure food safety.

Previous studies have shown that there is a delay in seroconversion in swine at low infective doses [15]. Our results also confirmed that when the infective dose was reduced from 600 to 200 larvae per pig, the load of larvae in diaphragm tissues reduced 10-fold (S3 Table), and continuous serological positivity was delayed by 60 days using ES-iELISA and by 55 days using rCLP-cELISA. It is worth noting that the antibody dynamic curve of CLP was completely different from that of ES at low infective doses. In rCLP-cELISA, the abundance of specific antibodies against CLP shifted below the threshold at 19 dpi up to 35 dpi (600 larvae) or to 60 dpi (400 larvae). Considering the delay time in antibody production for hosts, the deficiency of specific antibodies might be caused by the lack of CLP in newborn larvae with inadequate CLP expression in adult worms [22]. Moreover, the absence of persistent increases in CLP-associated antibodies resulted from parasitic immune status of the host [34], such as immunosuppression [35–37] or immune tolerance [38,39] in *T. spiralis* intestinal infection stage, to restrict the production of specific antibodies. In addition, a similar pattern of the antibody dynamic curve of CLP was demonstrated in mice in different dose groups. In the 400 larvae group, the level of specific antibodies against CLP remained above the threshold, which was consistent with our previous findings in swine at 50000 and 1000 larvae infections [23]. As detected in swine samples, CLP antibody levels in 10 and 20 larvae groups in mice also tended to change from above the threshold to below and then to consistently positive.

Although scientists have debated whether rodents are accidental hosts or stable reservoirs in livestock circulation (S4 Table), it is undeniable that rodents play an important role in the transmission of *Trichinella* and are also an important route for swine infection [9–12]. *T. spiralis* (T1) originating from Eastern Asia is now distributed world-wide due to its highly infective for domestic and sylvatic swine, mice and rat [1]. The risk of *Trichinella* infection in pig herds is presented mainly by *T. spiralis* (T1), representing a significant public health risk especially in Asia [1,7,8]. The proposed rCLP-cELISA was able to test specific antibody in mice infected with 3 isolates of *T. spiralis* (T1) and performed well in sera-diagnosis of *T. nelsoni* (T7) and *Trichinella* T8 infection. The results confirmed that the proposed rCLP-cELISA had the potential for cross-species diagnosis of *Trichinella* infection. For the twelve species/genotypes, the inconsistencies in the results of CWE-WB and rCLP-cELISA revealed differences in the antigenicity of the same epitope recognized by MAb 1H9 between these isolates. In addition, two different isolates of the same species, *T. murrelli* (T5), occasionally showed the presence or absence of the epitope (**Fig 4B** lanes 9–10 and S2 Text). To date, three proteins belonging to the cystatin superfamily have been reported, including CLP, muti-cystatin-like domain protein 1 (MCD-1) [40] and *T. spiralis* novel cystatin (TsCstN) [41], with no report on serological diagnosis-related applications about other two proteins. Epitopes recognized by MAb 1H9 were confirmed in 11 isolates, but the size and abundance of these bands varied from each other, suggesting the possibility of the presence of novel proteins that may or may not belonged to the cystatin superfamily. And in terms of inhibition rate, there are great differences in antigenicity between them.

Most of the data in the specificity study of ES-iELISA are from swine [14,15], ES-iELISA as well as rCLP-cELISA has good selectivity in the sera-diagnosis of swine infection. In contrast to rCLP-cELISA, ES-iELISA had cross-reactions in the serological detection of *Clonorchis sinensis* patients (**Figs 5D and 6E and 6F,** and S3). But rCLP-cELISA showed 100% specificity in the detection of patients with enterovirus and other parasites infections. The two samples that cross-reacted with ES-iELISA were tested by ES-WB, and bands of similar size to trichinellosis positive sera were found (**Figs 5D** and **6F**) with no bands found in rCLP-WB for corresponding samples (**Fig 6E**). Besides, all of the 8 samples from patients with *Clonorchis sinensis* showed higher OD values than other 28 samples (S3 Fig), although values of only 2/8 samples in *Clonorchis sinensis* group exceeded the threshold. Moreover, the two groups of samples were from two provinces with relatively far geographical locations, local residents had different dietary habits, and the negative serum had different band reactions in ES-WB (**Figs 5D** lanes 8–9 and **6F** lane 2). Thus, it was believed that partly similar band patterns can be caused by the reaction between ES and *Clonorchis sinensis* infected human sera. However, rCLP-cELISA showed better selectivity than ES-iELISA with PI ratios lower than 40% in these samples. Besides, the ROC analysis of 100 negative and 57 positive serum samples showed that rCLP-cELISA can distinguish between the presence and absence of *Trichinella* antibodies with considerable specificity (100%) and sensitivity (96.49%). Therefore, the proposed rCLP-cELISA has great potential in sera-diagnosis of human trichinellosis.

In conclusion, a competition ELISA employing the CLP and MAb was proposed and evaluated using positive sera from multiple *T. spiralis* infected hosts. Compared to ES-iELISA, the proposed rCLP-cELISA showed a similar positive detection rate in swine at low infection doses, with the seroconversion slightly delayed. In addition, the rCLP-cELISA exhibited good versatility in mice infected with different species/genotypes of *Trichinella*, and can distinguish the status of trichinellosis patients from healthy donors with considerable sensitivity and specificity. Taking advantage of its controllable quality stability, the rCLP-cELISA had potential applications for *T. spiralis*-related serological investigation and screening for multihost serum samples. With the development of MAb modification strategies [42,43] and the discovery of

antigenic proteins from ES products [20,44], the proposed rCLP-cELISA provides useful reference for the improvement of serological assay for monitoring the exposure of *T. spiralis*.

## Supporting information

**S1 Fig. PI ratios of 270 negative serum samples of backyard pigs in the rCLP-cELISA.** The average PI (X) of the 270 swine serum samples by the rCLP-cELISA was 33.14% ± 9.27% (mean ± SD), where X + 2 SD = 51.68%, and PI (%) ≥ 52% was considered positive. PI (%) < 52%, the serum antibody value was considered negative.
(TIF)

**S2 Fig. PI ratios of 30 negative mouse sera.** The average PI (X) of the 30 mouse serum samples by the rCLP-cELISA was 27.52% ± 5.76% (mean ± SD), where X + 2 SD = 39.04%, and PI (%) ≥ 39% was considered positive. PI (%) < 39%, the serum antibody value was considered negative.
(TIF)

**S3 Fig. Cross-reactivity testing of the ES-iELISA in human serum samples.** The procedures were implemented according to the manufacturer's instructions (Abcam, UK). Group a: Human sera of enterovirus (n = 6) and other parasites (n = 22) infections; Group b: Human sera of *Clonorchis sinensis* (n = 8) infections, two of the samples (green box) were tested positive by ES-iELISA; Group c: Substrate blank controls; Group d: Positive controls; Group e: Cut-off controls; Group f: Negative controls.
(TIF)

**S4 Fig. SDS-PAGE with silver dyeing of ES products of *T. spiralis* (T1, iss 534) muscle larvae.**
(TIF)

**S1 Text. Selection of MAbs bingding to native epitopes of CLP. Fig A The negative-to-positive ratios of the three MAbs at different dilutions under optimal conditions.** The negative-to-positive ratio of 1H9 was higher than those of 6B5 and 7F8 MAbs. The 1H9 was able to compete with the serum that contained antibody against *T. spiralis* for the available epitopes of rCLP antigens and was suitable for the development of rCLP-cELISA.
(DOC)

**S2 Text. Microscopy and genotying test in *T. murrelli* (T5, iss 415). Fig A Diaphragm microscopy for *T. murrelli* (T5, iss 415) infected mice (100×).** The black arrow showed the cyst of *T. murrelli* (T5, iss 415) larvae. **Fig B Genotyping test for two isolates of *T. murrelli* (iss 415 and iss 35).** Lane 1: DL2000 DNA marker; Lane 2: Products for DNA extraction from pooled larvae of *T. spiralis* (T1, iss 534); Lane 3: Products for DNA extraction from pooled larvae of *T. pseudospiralis* (T4, iss 141); Lane 4: Products for DNA extraction from pooled larvae of *T. murrelli* (T5, iss 35); Lane 5: Products for DNA extraction from pooled larvae of *T. murrelli* (T5, iss 415).
(DOC)

**S1 Table. Comparison of rCLP-cELISA with ES-iELISA and artificial digestion method in experimentally infected swine.**
(DOC)

**S2 Table. Performance of ES-iELISA for detecting experimentally infected swine induced by various doses of *T. spiralis*.**
(DOC)

**S3 Table. Larvae burden in diaphragm of *T. spiralis* (iss 534) infected swine using artificial digestion.**
(DOC)

**S4 Table. Prevalence of *T. spiralis* in rats trapped on pig farms with active transmission of *T. spiralis*.**
(DOC)

## Author Contributions

**Conceptualization:** Yan Liu, Ning Xu, Mingyuan Liu, Xiaolei Liu, Yu Zhou.

**Formal analysis:** Yan Liu, Ning Xu.

**Funding acquisition:** Mingyuan Liu, Xiaolei Liu, Yu Zhou.

**Investigation:** Yan Liu, Ning Xu.

**Methodology:** Yan Liu, Ning Xu, Yansong Li, Bin Tang.

**Supervision:** Yansong Li, Yu Zhou.

**Validation:** Bin Tang, Hualin Yang, Weihua Gao.

**Writing – original draft:** Yan Liu.

**Writing – review & editing:** Ning Xu, Yansong Li, Bin Tang, Hualin Yang, Weihua Gao, Mingyuan Liu, Xiaolei Liu, Yu Zhou.

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
