## [Decision Letter · Decision Letter 0]

14 Jul 2021

Dear Dr. Liu,

Thank you very much for submitting your manuscript "Recombinant cystatin-like protein-based competition ELISA for Trichinella spiralis antibody test in multihost sera" for consideration at PLOS Neglected Tropical Diseases. As with all papers reviewed by the journal, your manuscript was reviewed by members of the editorial board and by several independent reviewers. In light of the reviews (below this email), we would like to invite the resubmission of a significantly-revised version that takes into account the reviewers' comments. 

We cannot make any decision about publication until we have seen the revised manuscript and your response to the reviewers' comments. Your revised manuscript is also likely to be sent to reviewers for further evaluation.

Sincerely,

Subash Babu

Associate Editor

Sara Lustigman

Deputy Editor

Reviewer's Responses to Questions

**Key Review Criteria Required for Acceptance?**

**Methods**

-Are the objectives of the study clearly articulated with a clear testable hypothesis stated?

-Is the study design appropriate to address the stated objectives?

-Is the population clearly described and appropriate for the hypothesis being tested?

-Is the sample size sufficient to ensure adequate power to address the hypothesis being tested?

-Were correct statistical analysis used to support conclusions?

-Are there concerns about ethical or regulatory requirements being met?

Reviewer #1: -Are the objectives of the study clearly articulated with a clear testable hypothesis stated? YES

-Is the study design appropriate to address the stated objectives? YES

-Is the population clearly described and appropriate for the hypothesis being tested? YES

-Is the sample size sufficient to ensure adequate power to address the hypothesis being tested? YES

-Were correct statistical analysis used to support conclusions? YES

-Are there concerns about ethical or regulatory requirements being met? NO

Reviewer #2: The objectives of the study are clearly articulated and the study design is appropriate. The statistical analysis was correctly used.

**Results**

-Does the analysis presented match the analysis plan?

-Are the results clearly and completely presented?

-Are the figures (Tables, Images) of sufficient quality for clarity?

Reviewer #1: -Does the analysis presented match the analysis plan? YES

-Are the results clearly and completely presented? YES

-Are the figures (Tables, Images) of sufficient quality for clarity? NO

Reviewer #2: The presented analysis matched the analysis plan and the results are clearly and completely presented.

**Conclusions**

-Are the conclusions supported by the data presented?

-Are the limitations of analysis clearly described?

-Do the authors discuss how these data can be helpful to advance our understanding of the topic under study?

-Is public health relevance addressed?

Reviewer #1: -Are the conclusions supported by the data presented?YES

-Are the limitations of analysis clearly described?NO

-Do the authors discuss how these data can be helpful to advance our understanding of the topic under study? YES

-Is public health relevance addressed?YES

Reviewer #2: The conclusions are supported by the data presented and the authors discussed how these data can be helpful to advance our understanding of the topic under study.

**Editorial and Data Presentation Modifications?**

Reviewer #1: (No Response)

Reviewer #2: (No Response)

**Summary and General Comments**

Reviewer #1: The Msc deals with the comparison of the ES-iELISA (actual gold standard) and the newly in house developed rCLP-cELISA for the detection of Ab against Trichinella.

This research is important in the field and better diagnostic tools for Trichinella detection are needed.

I have a few remarks and comments, see below:

-Why authors choose to focus only on the specificity by infected pork with only three low doses?

-Figure 3: 

*I suggest to change for curves, the differents dot are not easy to read.

*To me, on this figure, the detection for the 200 larvae group is ok at J17 but not at J21. So I think the detection is at J30. But do authors have data between J21 and J30? If no, according to these results, we can consider the detection at J30.

-Figure 4:

* Legends of A and B at the same time is not easy

*What are the rounds? Error bars cannot be seen.

-Discussion:

* First paragraph is introduction and should be deleted

*Lines 370-371: To me decreasing from 600 to 200 is not a strong reduction in porks. So it is logical to have the same results.

Reviewer #2: The manuscript titled " Recombinant cystatin-like protein-based competition ELISA for Trichinella spiralis antibody test in multihost sera " by Liu and his colleagues developed a novel and sensitive cELISA for the detection of Trichinella spiralis infection. The research is of interest for the relative studies. 

However, some questions are required to improve in the present form.

1.There are some grammatical and spelling errors in the manuscript. For example, in line 33, “Conclusios” should be “Conclusions”; in lines 165 “2.0 M H2SO4” should be “0.2 M H2SO4”; in line 414 “ clonorchis sinensis” should be “Clonorchis sinensis”;

PLOS authors have the option to publish the peer review history of their article (what does this mean?). If published, this will include your full peer review and any attached files.

Reviewer #1: No

Reviewer #2: No
---

## [Editor Report · Decision Letter 1]

11 Aug 2021

Dear Dr. Liu,

We are pleased to inform you that your manuscript 'Recombinant cystatin-like protein-based competition ELISA for Trichinella spiralis antibody test in multihost sera' has been provisionally accepted for publication in PLOS Neglected Tropical Diseases.

Best regards,

Subash Babu

Associate Editor

Sara Lustigman

Deputy Editor

---

## [Editor Report · Acceptance letter]

20 Aug 2021

Dear Dr. Liu,

We are delighted to inform you that your manuscript, "Recombinant cystatin-like protein-based competition ELISA for Trichinella spiralis antibody test in multihost sera," has been formally accepted for publication in PLOS Neglected Tropical Diseases.

Best regards,

Shaden Kamhawi

co-Editor-in-Chief

Paul Brindley

co-Editor-in-Chief
